# Extracting medicinal chemistry intuition via preference machine learning

Oh-Hyeon Choung [1], Riccardo Vianello[1], Marwin Segler[2], Nikolaus Stiefl [1] ✉ & José Jiménez-Luna [2] ✉

The lead optimization process in drug discovery campaigns is an arduous endeavour where the input of many medicinal chemists is weighed in order to reach a desired molecular property profile. Building the expertise to successfully drive such projects collaboratively is a very time-consuming process that typically spans many years within a chemist's career. In this work we aim to replicate this process by applying artificial intelligence learning-to-rank techniques on feedback that was obtained from 35 chemists at Novartis over the course of several months. We exemplify the usefulness of the learned proxies in routine tasks such as compound prioritization, motif rationalization, and biased de novo drug design. Annotated response data is provided, and developed models and code made available through a permissive open-source license.

Drug discovery is a complex, multi-step process that operates at the interface between many chemical and biological sub-disciplines. In many stages of the pipeline, and specifically during lead optimization, medicinal chemists—wet-lab or computational—play a central role, as they are routinely tasked with identifying which compounds to synthesize and evaluate over subsequent rounds of optimization[1]. In order to do this, medicinal chemists often review data that includes compound properties such as activity, ADMET[2], or target structural information, among many others. Therefore, for a campaign to be successful it needs not only rely on the quality of the generated experimental data, but ultimately also on the robustness and soundness of the decisions made by the medicinal chemistry team working on it[3].

During their professional careers, medicinal chemists build an expertise that enables them to make their decisions (e.g., compound prioritization) more efficiently[4]. That is, they develop an intuition on the factors relevant for a compound to be successful on following iterations of the early drug discovery process. While attempts have been previously made to formalize such knowledge with rule-based approaches (e.g., structural alerts), or simple cheminformatics desirability scores (e.g., drug-likeness), capturing the subtleties and intricacies involved in the ranking ability of chemists remains a fundamental challenge. With that motivation in mind, in this work we

investigate whether part of this knowledge can be distilled into machine learning models. Such models can potentially then be deployed as an aid in during the decision-making process in lead optimization or other parts of the drug discovery pipeline, similar to other recommendation systems already reported in the industry[5–7].

Since medicinal chemistry is currently mostly a human endeavour, it is also inevitably prone to subjective biases[8]. Several studies[9,10] have evaluated to what degree medicinal chemists tend to agree on their own and the decisions made by their colleagues. Most tasks explored in these works included presenting chemists with a list of compounds to filter over several rounds, in order to evaluate whether their choices overlapped with those of their peers, and if they were self-consistent with their own prior selections. These studies reported overall a weak agreement between and within each chemist—the disparity in these results being associated to several psychological factors, such as loss aversion[11]. Another study[12], closer in nature to what we present in this work, evaluated whether a small group of chemists could rate compounds according to properties such as drug-likeness and synthetic accessibility via the use of a Likert-type scale[13], to then train a classical machine learning model on the obtained responses. A more recent study by Merck[14], used the same scaling strategy to model an in-house crowdsourced proxy for molecular complexity. While varying low to fair correlation degrees were found between the scores

[1]Novartis Institutes for Biomedical Research, 4002 Basel, Switzerland. [2]Microsoft Research AI4Science, CB1 2FB Cambridge, UK.
✉e-mail: nikolaus.stiefl@novartis.com; jjimenezluna@microsoft.com

assigned by the chemists in the previous two studies, the reported study designs could have been prone to the anchoring psychological effect, in which decisions are affected by subject- and situation-specific reference values[11]. A recent work with a similar experimental setup was also described in the context of the design of porous organic cages[15].

In this study we set to overcome those limitations by adopting a strategy that is well-known in the context of multiplayer games. We cast the goal of ranking a set of molecules as a preference learning problem and show that individual preferences can be captured via pairwise comparisons with a simple neural network architecture. A basic schematic summarizing the idea behind the study is provided on Fig. 1. Proof-of-concept data collection rounds were carried out to evaluate whether the proposed study design and methodology successfully overcame cognitive bias limitations that were present in previous studies. 35 (wet-lab, computational, and analytical) chemists at Novartis participated in the study, with over 5000 annotations collected over several rounds driven by an active learning approach. We show that the learned implicit scoring functions capture aspects of chemistry currently not covered by other in silico chemoinformatics metrics and rule sets, some of them derived from highly optimised internal annotations over years of cumulative know-how. We furthermore exemplify their applicability in the context of hit-to-lead compound prioritization and biased de novo machine-learning drug design. We also show that the proposed learned scoring function can capture the concept of drug-likeness more accurately than another widely used metric (QED). We furthermore rationalize the learned chemical preferences by means of fragment analyses on a large public compound database. Finally, so as to facilitate reproducibility and foster additional research on this topic, a software package (MolSkill), containing production-ready models and anonymized response data, is made available through a permissive license in an accompanying code repository.

## Results

We first focus on the evaluation of the results provided by two preliminary rounds for the study (see Methods), which ultimately led us to pursue the subsequent production-level runs. This is followed by a quantitative evaluation of predictive model performance over the production rounds. We then proceed to explore several areas where we believe the proposed scoring function can be practical. We study the relationship of the learned scoring function to other common in silico metrics in chemoinformatics and evaluate whether it can distinguish between chemical sets of different nature. We further investigate whether more precise learned chemical preferences can be rationalized via means of a fragment analysis and, finally, exemplify the usage of the proposed scoring function in biased molecular generation.

### Preliminary analysis rounds

Results for the two preliminary rounds are summarized in Table 1. As a measure for inter-rater agreement, we consider the Fleiss' $\kappa_F$ coefficient[16] among the responses provided by the chemists in both preliminary rounds. We measured $\kappa_{F1} = 0.4$ and $\kappa_{F2} = 0.32$ for the first and second round, respectively, and concluded that there was a moderate agreement between the preferences expressed by the chemists. One likely reason for the observed level of agreement is the fact, that especially in cases where there was no clear-cut preference, decisions were driven by prior personal experiences. Still, these results suggested that there was a pattern to be learned by the responses to the posed question. Using the redundant pairs present in both preliminary rounds, we also evaluated per-chemist intra-rater agreement using the Cohen's $\kappa_C$ coefficient. With $\kappa_{C1} = 0.6$ and $\kappa_{C2} = 0.59$ for the first and second preliminary round, respectively, we conclude that that in most cases, chemists displayed a fair degree of response consistency. In addition, no specific positional bias on the screen where the questions were posed was detected for any of the preliminary participants, with preferences reasonably close to the expected random 50% baseline. Additional two-by-two inter-rater agreement coefficients are presented in Fig. S1, from which we draw similar conclusions.

Overall, the results on the preliminary rounds suggested that there was indeed a signal to be learned from the opinions expressed by the chemists that had participated in the study up to that point. These findings convinced us to extend the study and continue with the subsequently presented, larger production-level runs.

### Predictive pair preference performance

In order to evaluate whether the trained model had successfully learned the preferences expressed by the chemists, we iteratively measured its predictive performance via the area under the receiver-operating characteristic (AUROC) curve under different scenarios (Fig. 2). Specifically, we kept the data from the preliminary rounds as external sets for validation that are not used for model training or uncertainty quantification during the active learning rounds. In

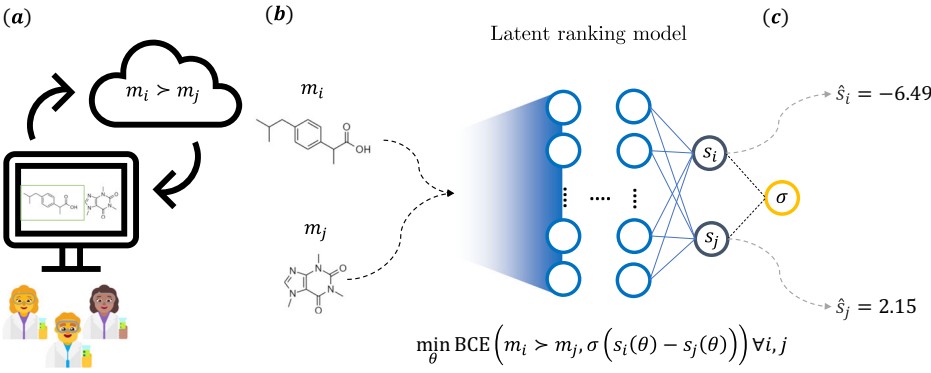

$$\min_{\theta} \mathrm{BCE}\left(m_i \succ m_j, \sigma\left(s_i(\theta) - s_j(\theta)\right)\right) \forall i, j$$

**Fig. 1 | Overall schematic of the main idea behind the study. a** Molecules are treated as players in a competitive game, with the probability of one winning over the other provided by feedback supplied by chemists. For this, the chemists are asked to select one of two molecules upon presented with a pre-specified question prompt on a web application. **b** An implicit score model is learned based on this feedback. A two-legged feed-forward neural network with fixed weights in each leg is supplied with pairs of molecules featurized with common cheminformatics descriptors. During training, its parameters are optimized via a binary cross-entropy loss (BCE) loss that depends on a latent score difference computed on the molecule pairs and feedback supplied by the chemists. **c** Once trained, scores can be inferred for any arbitrary molecule, which can then be used for downstream cheminformatics tasks. Symbols: $s_i, s_j$: scores computed for molecules $m_i$ and $m_j$, respectively. $\sigma$: sigmoid function. $\theta$: model parameters.

**Table 1 | Intra-rater agreement, as measured by the percentage of times chemists agreed with their previous choice on a pair and by the Cohen's κ coefficient**

| Chemist Id. | Intra-rater Ag. (%) | | Intra-rater Ag. (Cohen's κ) | | Left-right bias (%) | |
|---|---|---|---|---|---|---|
| | R1 | R2 | R1 | R2 | R1 | R2 |
| 1 | 100.0 | 100.0 | 1*** | 1*** | 48.2 | 47.7 |
| 2 | 92.1 | 84.2 | 0.68*** | 0.37* | 47.2 | 54.5 |
| 3 | 86.8 | 78.9 | 0.49* | 0.16 | 48.6 | 55.5 |
| 4 | 79.8 | 84.2 | 0.27 | 0.35* | 54.6 | 48.2 |
| 5 | 85.1 | 92.1 | 0.37* | 0.69*** | 47.2 | 48.6 |
| 6 | 89.5 | – | 0.55** | – | 47.2 | – |
| 7 | 84.2 | 92.1 | 0.33 | 0.65*** | 48.6 | 47.7 |
| 8 | 94.7 | 92.1 | 0.79*** | 0.69*** | 46.8 | 51.8 |
| 9 | 95.6 | 89.5 | 0.89*** | 0.58*** | 50.9 | 48.2 |
| 10 | – | 81.6 | – | 0.28 | – | 52.7 |
| 11 | – | 92.1 | – | 0.69*** | – | 53.2 |
| 12 | – | 92.1 | – | 0.69*** | – | 56.8 |
| 13 | – | 92.1 | – | 0.69*** | – | 50.9 |
| 14 | – | 89.5 | – | 0.58** | – | 51.8 |
| 15 | – | 94.7 | – | 0.79*** | – | 54.1 |

Left-right bias measured as the percentage of times a rater chose the compound presented on one side of the screen.

$R1/R2$ First/second preliminary round of the study.

$***p < 0.01, **p < 0.05, *p < 0.1.$

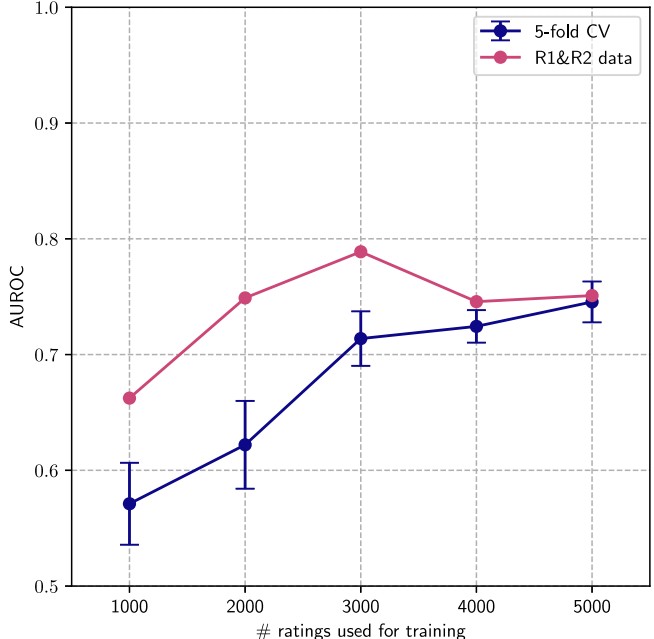

**Fig. 2 | Model benchmarking at different training set sizes.** Predictive performance of the proposed latent score ranking model when evaluating which compounds are preferred within each pair. Results presented at different train set sizes corresponding to the associated active learning batches considered during the study. AUROC area under the receiver-operating-characteristic curve, CV cross validation.

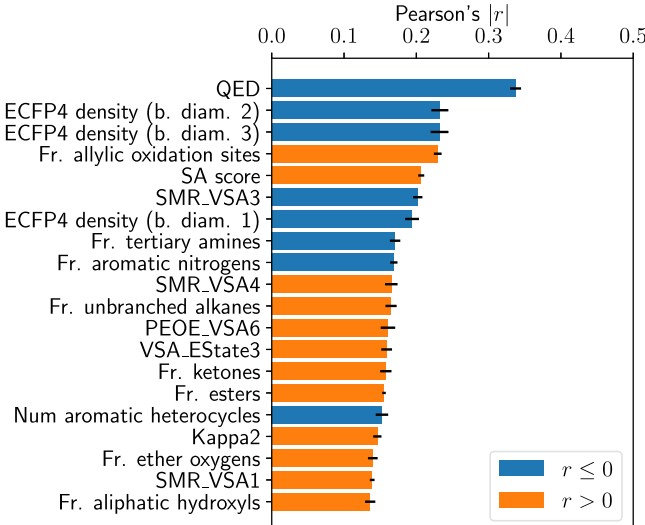

**Fig. 3 | Relationship between learned scores and other cheminformatics metrics.** Average absolute correlation coefficients (±1 standard deviation) between several in silico descriptors computed via RDKit and learned compound scores (lower is better) in the training set ($n = 5276$). Results shown for the 20 most correlated in silico metrics (in absolute value) over five different training seeds.

available pairs thresholds, respectively. Interestingly, cross-validation results did not display reaching a performance plateau even when evaluated at the last available batch of responses, hinting that performance could further be improved if more data had been collected. In addition, performance increased during the first three collected batches but stayed relatively stable around the 0.75 AUROC mark when models were evaluated on the preliminary round data, which could be explained by the limited amount of pairs available in these sets. Overall, these results suggest that the model is able to correctly learn preferences as expressed by medicinal chemists in the current experimental setup. For completeness, we also evaluated to what degree different common molecular representations had an impact on model overall performance (Fig. S2).

## Relationship to other in silico metrics

One of the main assumptions of the main question presented to the participants in this study is that, over the course of their careers, medicinal chemists develop an expertise that is hardly quantifiable by other existing in silico metrics. In order to evaluate whether such is the case, we measure to what degree the learned compound scores correlate with other ligand-based properties that are commonly used during optimization (e.g., drug-likeness, topological surface area, number of saturated rings). All properties considered were computed with the RDKit software package[17]. A summary of the highest correlated properties (on an absolute scale) in the training data is presented in Fig. 3. With Pearson correlation coefficients overall not surpassing the $r = 0.4$ threshold, we conclude that the learned scores are in fact providing a perspective on molecules that is orthogonal to what can be currently computed with other cheminformatics software routines. Among the most correlated properties we can find: drug-likeness[18], fingerprint density, the fraction of allylic oxidation sites, atomic contributions to the van der Waals surface area[19], or the Hall-Kier kappa value[20]. Not surprisingly, the most correlated descriptor in these analyses is QED, which also attempts to capture drug-likeness. Interestingly, the fact that different flavours of fingerprint density are also present within this list suggests that chemists seem to display a slight preference towards richer molecules feature-wise. To some degree this is not surprising, as one main example of features that would result on low fingerprint density are repeating motifs of similar atom types (e.g.,

addition, we also evaluated model performance via randomized five-fold cross-validation after each labelled batch of 1000 samples. From the cross-validation results, a steady pair classification performance improvement can be observed as more data became available, starting from 0.6 and surpassing 0.74 AUROC values at the 1000 and 5000

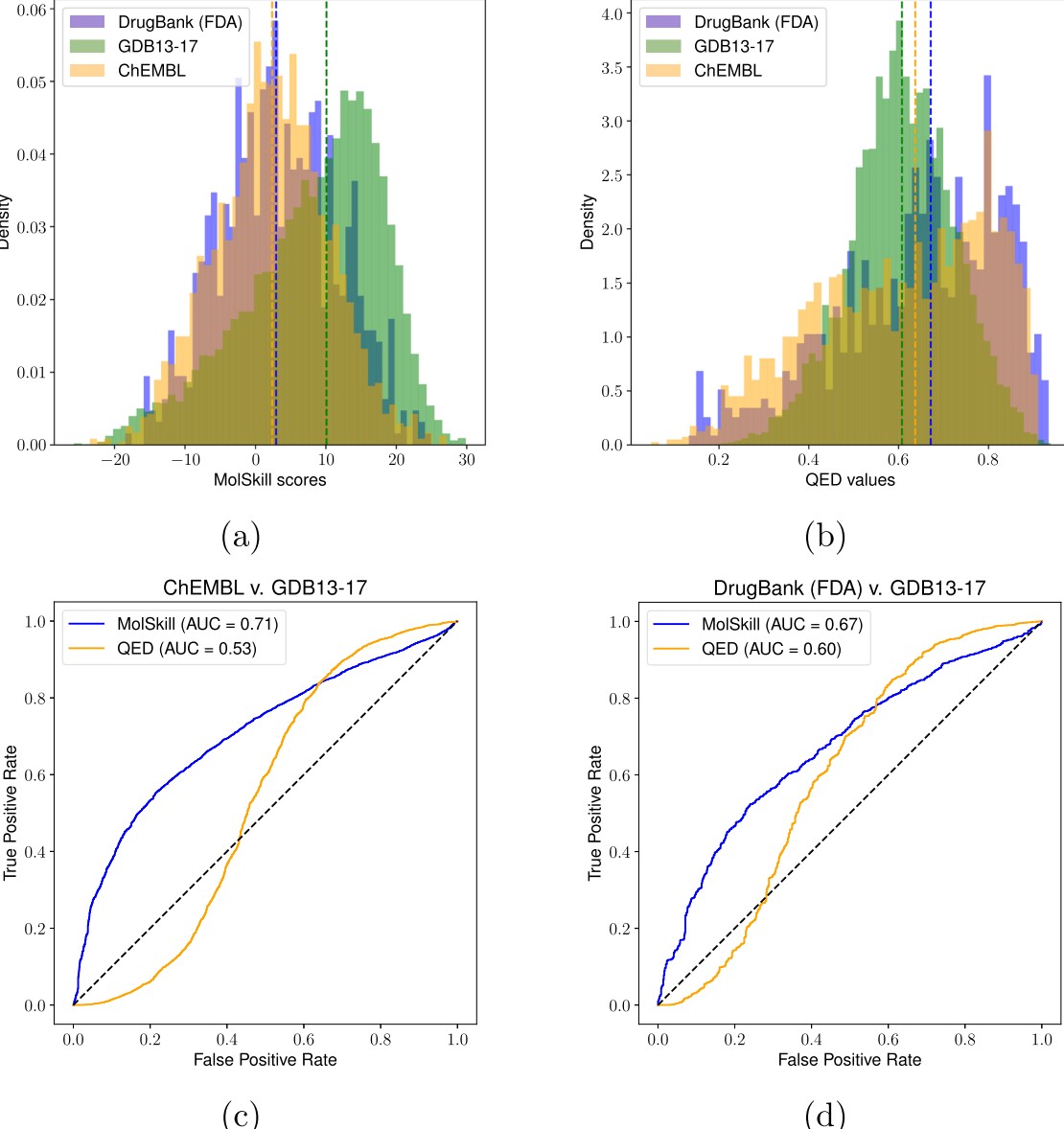

**Fig. 4 | Using the learned scores to discriminate between different chemical sets. a, b** Distribution of MolSkill scores and QED values over three different molecular sets: ChEMBL, a set of FDA-approved drugs as made available by Drug-Bank, and a random sample of the combinatorially generated GDB sets. **c, d** ROC AUC curves for both MolSkill scores and QED values when tasked to discriminate between molecules from either ChEMBL or FDA-approved drugs from GDB-extracted molecules.

long aliphatic chains), which are difficult to optimize or prone to overall unspecific binding. In contrast, a small positive correlation with the SA score measure[21] can be observed, which hints that the proposed score slightly prefers synthetically simpler compounds. Another noteworthy fact is that the SMR VSA3 descriptor, a measure of molecular surface area that is aggregated according to Wildman-Crippen MR value limits[22], is slightly correlated negatively. This could hint that chemists have a liking towards molecules that feature neutral nitrogen atoms. We however stress that the magnitude of these correlations is, in our opinion, insufficient to make any strong claims. Nevertheless, for completeness, an extensive list of all of the properties computed as well as their correlations to the learned scores is also provided in Fig. S3.

**Discriminating between chemical sets**
As a way of quantitatively evaluating whether the learned scores can be used to deprioritize compounds that could be seen as undesirable, we

consider an approach similar to one reported in the original QED study[18]. Specifically, we scored different sets of molecules: (i) a set of 2386 ChEMBL compounds not present in our training sets with at least an annotated maximum phase of development over 0.5, (ii) a set of FDA-approved drugs as made available by the DrugBank[23] database, and (iii) a random subset of 10,000 compounds extracted from each the GDB13 and GDB17 databases[24,25]. Furthermore, we used the latter GDB compounds as a control, since they were originally generated in a combinatorial fashion and should in practice contain molecules that do not exhibit drug-like properties. To ensure that the molecules considered in these analyses did not fall outside of the applicability domain of the trained model, we made sure to apply the same filtering procedures as those detailed in the Methods section (see Data retrieval, cleaning, and pair generation). This resulted on 732, and 8616 analyzed molecules for the FDA-approved drugs and GDB sets, respectively. As a baseline method to compare the learned scores with, we considered the standard QED implementation as available on the

RDKit package. On Fig. 4a, it can be observed that the distribution of learned scores is clearly well separated between sets better representing drug-like space (in other words those more apealing to medicinal chemists, i.e., Drugbank FDA-approved drugs and ChEMBL) against the GDB set. QED scores (Fig. 4b) on the other hand, struggle at making such separation between the three sets. While an one-way ANOVA test was performed and the null hypothesis of equal mean values between the three sets was rejected for both methods with virtually zero $p$-values ($F_{MolSkill}$ = 546.88, $F_{QED}$ = 22.83), receiver operating characteristic curves to distinguish the drug-like sets against GDB showed that only the proposed learned scores were predictive enough in practice for this task (Fig. 4c, d).

However, and while not explicitly trained to do so, it should be noted that one major factor on why the proposed scoring function is able to distinguish between these two sets could be related to overall molecular size. Specifically, the molecules in both the ChEMBL and FDA-approved sets are significantly larger than those present in the GDB sets. When adjusting for molecular size (i.e., filtering out molecules larger than a certain amount of heavy atoms in the drug-like sets), we can observe that QED can be used to efficiently discriminate between both ChEMBL and the FDA-approved from the control set (Fig. S4) at smaller molecular size ranges. For the ChEMBL set, we mainly attribute these differences to the fact that QED scores are very negatively correlated with the number of heavy atoms whereas MolSkill is not (Fig. S5). Performance dependence on molecular size for the FDA-approved drugs set is close to random for very small molecules (i.e., -17 heavy atoms) for both methods, albeit, as in the case for the ChEMBL molecules, MolSkill scores become more performant when considering molecules featuring more than 30 heavy atoms. Another remark is that some of the smaller molecules contained in the drug-like sets would be nowadays more accurately described as a fragment or a lead rather than a drug (e.g., acetylsalicylic acid).

## Exploring fragment preference

As means for model interpretability, in this section we aim to disentangle whether the learned scores exhibit a bias towards specific molecular motifs. In order to do so, we make use of the BRICS algorithm[26], as implemented in the RDKit software, and compute all available leaf fragments and associated model scores for each molecule present in the training set. Since the fragments contained an attachment atom type not seen during training, fragments were scored according to the average scores of the compounds they were substructures of in the training set. In addition, to avoid biases related to uncommon motifs or unexplored areas of chemical space, only fragments appearing a minimum of 5 times in the training set were considered in this analysis. A small selection of the highest and lowest ranked fragments is presented in Fig. 5. Among the worst-ranked fragments we can observe undesirable groups such as phenols, free acids, ketones, thioureas, allyls, long alkyl chains, naphtyls, cumarines, Hantzsch esters, quaternary amines, sugar-like structures or highly substituted rings. On the other hand, among the best-ranked groups we can find many commonly used medicinal chemistry motifs such as pyrazines, pyrimidines, sulfones, imidazoles, oxadiazoles, phenyls, or bicyclic heteroaromatics. Qualitatively, this experiment suggests that the proposed score has learned patterns that are in line with motifs present in existing drug-like molecules. The full set of fragments, their frequency in the data, as well as their associated MolSkill scores are provided in the accompanying code repository to this work.

## Biased molecular design

As a way of exemplifying how the implicitly learned scoring function may be applied in a realistic setting, in this section we use it to bias a generative model towards favourable regions of chemical space. We make use of the GuacaMol baselines[27] package and implemented a

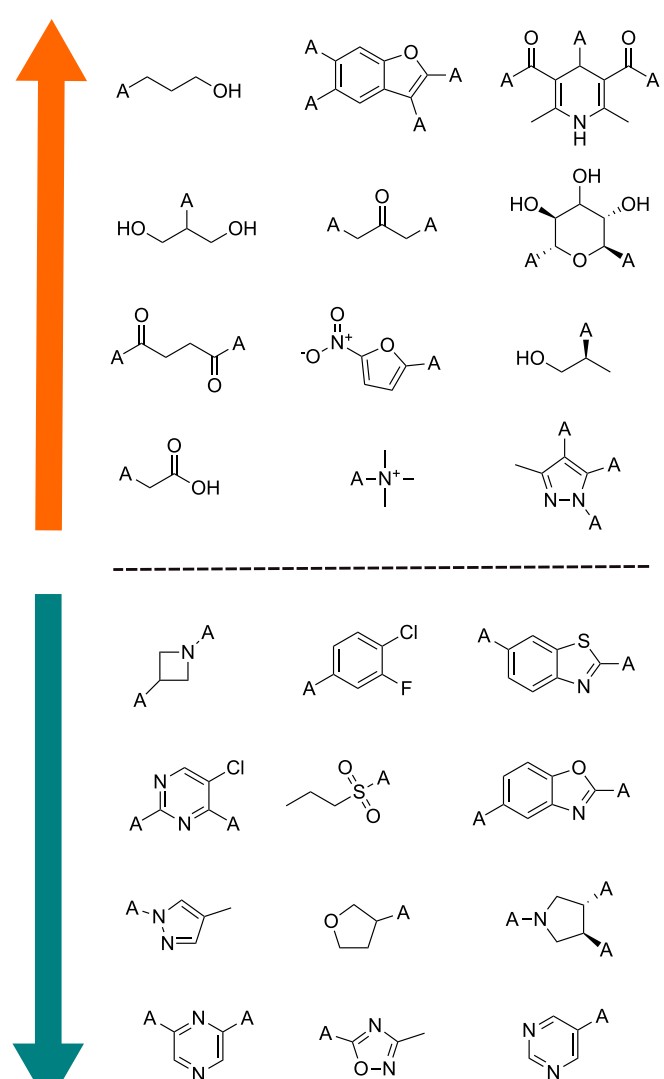

**Fig. 5 | Model fragment preference.** Some fragment examples evaluated by the learned scoring function. Fragments representative at each end of the score distribution (lower is better).

submodule with the proposed scoring function trained on all available rating data. We then chose the pretrained SMILES-based LSTM generative model and the hill-climbing optimization strategy[28] to generate 500 molecules both maximizing and minimizing the learned scoring function. Some generated molecule examples are presented in Fig. 6. Visually inspecting some of the examples maximized by the scoring function, we can appreciate that the model is assigning high (i.e., unfavorable) scores to compounds that feature long flexible chains, atypical groups such as phosphates or azides, conjugated double bonds, reactive pieces, or overall higher number of carboxylates and alcohols, among many other non-drug like properties. On the other hand, minimizing the learned scoring function results in a reasonable mix of aromatic rings and aliphatic $sp^3$ carbons, reasonably sized fragments as well as several typical groups featured in drug-like molecules. From these qualitative analyses we conclude that the scoring function has successfully captured a reasonable degree of chemical intuition.

One caveat that we had experimentally observed during molecular generation is that it was useful to constrain or stop optimization of the scoring function once it had reached values close to the limits of the empirical distribution of learned scores ($|\hat{s}| \approx 9$ using the reported regularization strategy applied during training in our sets). Not doing

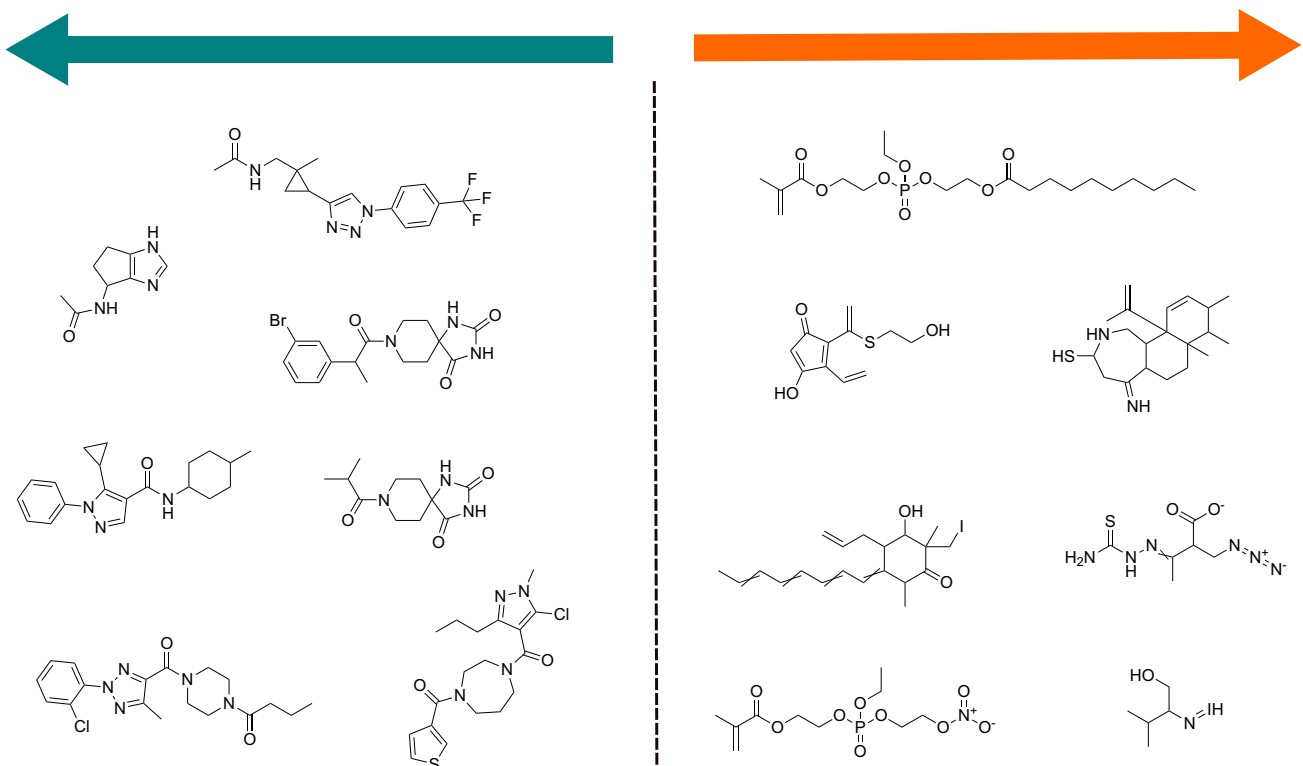

**Fig. 6 | Biased de novo design via predicted scores.** Some molecular examples prioritized by the proposed implicit scoring function when paired with a generative model. Results presented for both maximization and minimization of the learned score (lower is better).

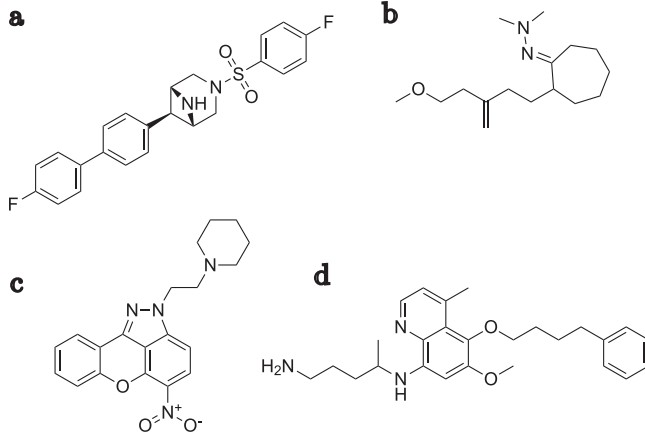

**Fig. 7 | Deprioritizing unwanted compounds with the score model.** Example compounds not flagged by the NIBR filters but effectively deprioritized by the learned scores (9 or higher). **a** Displays complex features, **b** shows a mix of flexibility and feature richness coupled with undesired distribution of features, **c** displays complex features, and **d** exhibits both flexibility and feature richness.

so resulted in a certain degree of quirkiness and molecular invalidity, which we attribute to the generative algorithm overexploiting the scoring function on regions of chemical space that it had not previously observed during training. Additional details on the generative model and optimization hyperparameters are made available in the accompanying code repository to this work.

**Qualitative score assessments on ChEMBL**
While the quality of the generated molecules indicate a high relevance of the proposed scoring function for de novo drug design, we

furthermore qualitatively evaluated its usefulness to filter out undesirable compounds. This was studied especially in the light of existing rule-based approaches, such as the NIBR filters[29], which are routinely used to deprioritize and flag problematic compounds before consideration. Ideally, our goal was to rationalize compound features not necessarily captured by such methods currently, and at the same time considered as undesirable by medicinal chemists. Towards this goal, we manually reviewed molecules from the initial pool, which had been already filtered with simple properties as well as with the aforementioned rules, and then visually inspected those that were assigned a high score by the proposed function. Figure 7 shows four of such compounds. While there are some features that could be described as unattractive and can be captured with a generic SMARTS pattern (e.g., the terminal alkene in compound *b*, or the aromatic nitro group in compound *c*), the overall unattractiveness seems to be driven by more general properties. Based on our subjective opinion, among others these seem to include compound complexity (*c* and partially *a*), a mix of flexibility and feature-richness (*b* and *d*), or the distribution of features (*b*). While theoretically possible, defining such rules explicitly is a difficult task and is unlikely to capture all undesirable cases.

## Discussion
In this work, we have described the development of a machine-learned scoring function of human preference in the context of early drug discovery campaigns. We have done so by adapting the well-known framework of player ratings to a pairwise learning-to-rank experimental design between molecules. In order to do so, we have internally deployed a large user study at Novartis, where the expertise of 35 medicinal chemists was taken into account. In more detail, in this study we show that such expertise can be successfully learned by a latent score machine-learning model. Such scores have been shown to be providing additional or orthogonal information to what can be obtained by other common in silico ligand-based properties or

substructure-based fragment definitions. We also exemplified the utility of such modelling approach in several routine cheminformatics tasks, such as the deprioritization of compounds currently not flagged by well-known rule-based approaches, or biased molecular design via a generative ML model. We furthermore rationalized and motivated what the model has learned by means of a fragment analysis on a large set of compounds and show that it outperforms a popular quantitative measure of drug-likeness at distinguishing chemical sets of different nature.

A main limitation of the study relates to the simplicity of the question asked during data collection, which was left intentionally vague to capture chemical intuition on a timely manner. In addition, while the proposed study design results in a higher agreement between participants than when compared to previous works, the pairwise comparison approach is not free from criticism. There is a growing body of evidence from behavioural psychology pointing that the cognitive bias research tradition[11], which claims that rationality is based on preference consistency, paints an incomplete picture of intelligence. Several authors[30,31] have instead pointed out that humans can flexibly behave in different contexts and can use emotions to achieve personal goals, which could be interpreted as rational cognitive biases[32]. It is also well known that humans have a tendency to simplify high-dimensional problems (such as the one proposed here, choosing a molecule out of a pair) into a handful of variables that they can keep track of cognitively (i.e., the so-called Flatland fallacy[33]), which likely depend on each individual chemist. Accounting for factors such as these is, to our knowledge, still an unexplored question in the field.

We see the utility of the proposed model to go beyond what is proposed in the current study. Specifically, we believe that there is potential to extend the discussed setup for other observables in drug discovery that are inherently quantifiable but expensive to obtain experimentally (e.g., compound stability calculations). In addition, we believe it could provide insights into unexplored regions of chemical space currently ignored when applying simpler mnemonics such as Lipinski's rule of five[34,35]. With that in mind, we believe that soft versions of some popular rule-based filters can be learned by artificially generating training pairs alongside a similar architecture as the one proposed. Such models could potentially overcome the main limitation of having to pre-filter compounds before inference so as to avoid out-of-distribution risks. Along those same lines, the proposed score can also be used to prioritize combinatorially generated chemical libraries, whose inherent novelty makes them hard to filter with existing rule-based methods. Another venue of further research is examining the utility of the study setup in prospective, target-specific lead optimization scenarios, where information from multiple sources (e.g., biological profiles, ADMET) need to be taken into account as a whole to successfully deliver a drug to the market.

Finally, from a hands-on experience from several ongoing medicinal chemistry projects at Novartis, MolSkill is currently being applied in several routine tasks. Specifically, in an era where machine-learning methods can design tens of thousands of compounds, or technologies such as high-throughput screening can highlight a large number of hits at early stages of the drug discovery process, the proposed score is being used to implicitly incorporate chemists' intuition for compound filtering without the requirement of manual examination. This usage will, hopefully, accelerate both the adoption and trust on generative approaches in the upcoming years.

## Methods

### User composition and question design
A total of 35 medicinal chemists from different sites at Novartis participated in the presented study. These included chemists from different geographical sites, at different levels of seniority/expertise, and from either a medicinal, organic, analytical, or computational chemistry background.

In regards to the question posed, and in the belief that chemists develop an inherent sense of what constitutes a desirable compound over their careers, we set out to present a fairly simple, and intentionally ambiguous prompt asking them which of two presented compounds they preferred. We asked chemists to imagine an early virtual screening campaign setting (accounting for simple aspects such as oral availability and small molecular profile, but no other modalities such as covalency or bifunctionality) where they needed to decide which compound to follow up between two. The question was designed so that participants did not spend a significant amount of time evaluating each presented pair of compounds, while being generic enough so that one of the compounds could be discarded according to a non-defined gut feeling chemical preference. This could include drug-likeness, synthetic accessibility, or other criteria inherent to the pair of the compounds presented in each choice. We note that the question choice can be seen as an oversimplification of the problem, and that in other drug discovery scenarios, additional details on the presented prompt would be needed for clarification. In real-life setups, these details would typically include aspects like existing ADMET or activity data, or bespoke predictive models for those endpoints.

### Evolution of the study
Over the course of the presented study, several rounds were conducted. Two preliminary analysis rounds consisting of 220 molecular pair evaluations, and with feedback requested from 9 and 14 chemists at Novartis, respectively, were carried out. Specifically, we mainly focused on measuring:

- To what degree the choices made by one chemist agree with those made by their peers (i.e., inter-rater agreement). This was evaluated with 200 different compound pairs. Intuitively this a direct measure of whether there is a signal to be learned by a machine-learning model.
- Whether chemists choices are self-consistent (i.e., intra-rater agreement). In order to do so, we included an additional redundant 20 compound pairs, albeit in a random order and position on the screen.

In addition, we also studied whether there was a bias towards choosing a compound depending on its position on the screen (left/right) during annotation. After the first initial preliminary round was completed, we had received qualitative feedback from the chemists on some of the presented pairs. Specifically some criticism was expressed in regards to some pairs being inherently hard, as both compounds contained clearly problematic motifs (e.g., plague *vs.* cholera pairs where both compounds featured known toxicophores). These were then removed (see Data retrieval, cleaning, and pair generation section for details on the protocol). A second round with identical number of pairs was subsequently carried out. Note that in the first and second preliminary rounds, all chemists were handed out the same pairs (i.e., we performed inter-rater repetitions), so as to adequately evaluate the points presented above. After both preliminary rounds had yielded satisfactory results, we set out for a production run where we obtained over 5000 responses over the course of several months. Furthermore, since a reasonable degree of agreement between the chemists in the preliminary rounds was observed, we forwent the pair repetition requirement in the production runs and considered all participating chemists as a single labelling oracle.

### Data retrieval, cleaning, and pair generation
For all purposes of the study, we use compounds extracted from the publicly available ChEMBL database[36] (version 31). Specifically, all compounds considered in this study come from a pool where the following filters were applied: their molecular weight was between 200 and 1000 g mol$^{-1}$, their drug likeness (QED)[18] between 0.2 and 0.9,

and allowing up to 2 rule-of-five violations[37]. In addition, all retrieved compounds were checked so that they could successfully be read by the RDKit package[17], and subsequently standardized, which included removal of salts, tautomer normalization[38], and atom neutralization via O'Boyle's nocharge code[39]. For the second preliminary study round and subsequent production rounds, the NIBR substructure filters were also applied[29], which resulted in a final pool of 1,831,052 molecules.

For the two preliminary study rounds, and for the first round of the production stage, compounds present in the initial pool were grouped in 1000 clusters via the $k$-means algorithm, as implemented in scikit-learn[40], and using binary extended-connectivity fingerprints as molecular features. Pairs were then selected by ensuring that their associated clusters were not repeated within the same batch of questions. In addition, pairs with both compounds featuring either more than 10 rotatable bonds or 3 fused rings were removed. This was done to avoid comparisons where chemists may reject one compound or the other without a clear preference for either. We note that these filters were mostly informed by early informal discussions with the participating chemists, but are fairly arbitrary−less stringent boundaries could be chosen for future studies.

## Psychometric study setup

We considered a user study where interviewees were presented with pairs of choices (i.e., compounds) to select from. There were several reasons to consider a pairwise experimental design in contrast to simpler alternatives such as obtaining direct feedback on individual samples. One of such advantages is that there is abundant evidence from psychometric studies and decision theory suggesting that humans find it inherently hard to sort items according to their preferences[41], whereas making binary decisions is a task that is in general considered easier[42–44]. In addition, it also avoids user or situation-specific baseline biases: humans are known to start labelling from an anchor value that is then adjusted towards a final decision in situations of uncertainty or stress, which had been demonstrated to be an issue in other user studies[45–49]. In the specific context of drug discovery applications, it should be observed that this labeling strategy is markedly different from ones used in previous studies[14], where it was found that a large number of repetitions per compound was crucial to obtain a reliable proxy for the crowd-sourced endpoint of interest. This was mostly due to the overall low reported agreement between chemists, as also confirmed by another studies[10]. We partially attribute this difference in agreement to the human anchoring effect when rating items in a sequential manner.

## Learning to rank

Our setting resembles that of preference learning by pairwise comparisons[50]. One naïve approach to tackle the challenges raised by the proposed pairwise design is to try and induce a utility function based on how many times a compound has been preferred over others (or its proportion), and then frame this problem as a regular supervised regression task. The main disadvantage of this procedure, however, is that it requires the same compound to be present in several comparisons in order to accurately estimate a preference, which severely limits how much chemical space we can explore given a finite amount of time provided by the volunteer chemists. Instead, we take inspiration from the ELO skill-based systems that were popularized by the rating schemas for zero-sum games such as chess or backgammon[51], or more recently by the TrueSkill algorithm[52,53] as used by the Xbox Live multiplayer videogame service. In the original setting, the difference in ratings between two players served as a function of the probability of one player winning over the other. In our case, we consider the presented molecules to the chemists as the players participating in our game, the main goal being to rank them[54].

Mathematically, given a (possibly incomplete) set of molecules $m_1, m_2, \ldots, m_n \in \mathcal{M}$, and training data consisting of $k$ pairs of examples with binary preference relations of the type $m_i \succ m_j$ (meaning that $m_i$ was preferred over $m_j$ in a specific match), our task is to infer a total ordering over all molecules in $\mathcal{M}$. Furthermore, such pairs do not need to specify a complete ranking of the training data or be consistent (i.e., satisfy transitivity). In order to do so, we consider a function $s : \mathcal{M} \to \mathbb{R}$, where we assume that each molecule can be parameterized by a latent score that can be learned by a sufficiently expressive model[55]. Once this function has been approximated, it can be then used to impose a complete order over already seen or new molecules. Denoting by $\delta_{ij} := \hat{s}(m_i) - \hat{s}(m_j)$ the learned latent score difference between molecules $m_i$ and $m_j$, we estimate $\hat{p}\left(m_i \succ m_j\right) := \sigma\left(\delta_{ij}\right)$, where $\sigma$ is a sigmoid function. To learn $s$, we then simply use standard stochastic gradient descent and minimize a binary cross-entropy loss between the probability estimates and the preference values in the training data. Since this loss is a function of the learned $\boldsymbol{\delta}$ values only, to ensure identifiability of the scores $\boldsymbol{s}$, and to guarantee that these are centered around the real origin, we use a regularization term $\mathcal{L}_{\text{reg}}(\hat{\boldsymbol{s}}; \lambda) := \lambda |\hat{\boldsymbol{s}}|^2$, where $|\cdot|$ is the Euclidean norm and $\lambda$ is a user-defined hyperparameter. Empirically, we found that setting small values $\lambda \simeq 10^{-6}$ is enough to encourage the desired score behaviour for our use case.

We chose to parameterize $s$ as a standard feedforward neural network that uses 2048-bit count-based extended connectivity fingerprints[56] and a list of two-dimensional descriptors computed via RDKit as input features. We train all models using the Adam[57] optimizer with an initial learning rate of $3 \times 10^{-4}$. Additional molecular featurization and architectural details are available in the accompanying code repository to this study.

It is important to note that the proposed methodology is markedly different in design from previous studies, which directly model the human-provided feedback as a regression objective. Our approach, on the other hand, infers these scores automatically based on the preferences expressed, which, due to the study design are more robust between and within chemists. This in practice allows us to prioritize the exploration of a wider chemical space rather than having to rely on repeated feedback from different chemists.

## Active learning

To achieve the set goal of 5000 user responses in the study, and to ensure we covered sufficient chemical space, we considered a simple batched active learning approach[58,59]. Specifically, every 1000 responses we randomly sampled a large number of pairs from the initial pool of compounds. These pairs were then ranked according to their uncertainty, as estimated by the variance of their predicted $\delta_{ij}$ values using the Monte Carlo dropout method[60] with a fixed rate of 0.2 and 100 predicted samples. In addition, to ensure that comparisons were not drawn between too many compounds belonging to similar regions of chemical space, we used a clustering strategy (see Data retrieval, cleaning, and pair generation section) and allowed up to one comparison between any two clusters in each batch.

## Platform deployment

A platform for questionnaire delivery was internally developed at Novartis. Users were asked to select between pairs of compounds presented upon a predefined question. The front-end was developed using an intuitive ReactJS (reactjs.org) web GUI that could be operated either via a computer or a touchscreen device. A screenshot of the deployed interface is shown on Fig. 8. Special care was taken to ensure that the same pair was not presented to different users. Results were internally stored in a remote PostgreSQL database[61] instance through a custom REST API developed with FastAPI (fastapi.tiangolo.com). The database was then periodically exported to perform model training and run analyses.

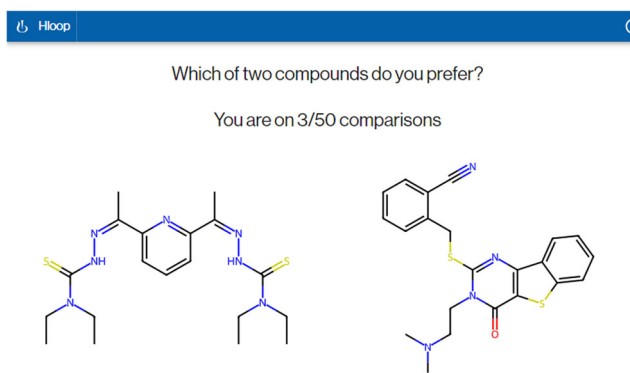

**Fig. 8 | Screenshot of the web interface used for data collection during the study.** Chemists were asked to select which of two compounds they preferred according to a prespecified question presented at the top of the page.

### Reporting summary

Further information on research design is available in the Nature Portfolio Reporting Summary linked to this article.

### Data availability

All data generated in this study, including survey responses as well as training and generated molecules have been deposited in Zenodo under accession code https://doi.org/10.5281/zenodo.8214903[62]. Participants in the study opted-in and informed consent was obtained from all participating chemists by Novartis.

### Code availability

Production-ready trained models and all related code are made available via a MIT-licensed repository github.com/microsoft/molskill[62]. A conda package is also provided for integration convenience within downstream cheminformatics tasks. Neural network models were trained using the PyTorch automatic differentiation library (version 1.11).

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

## Acknowledgements
We are grateful for the feedback (and criticism!) received from all the participant chemists at Novartis, without whom this study would not have been possible. We also thank G. Landrum, P. Walters, and the entire GenChem teams at both Novartis and Microsoft Research for helpful discussions on this work.

## Author contributions
O.C.: platform deployment, analysis, manuscript writing; R.V.: platform deployment; M.S.: conceptualization, manuscript writing; J.J-L.: super-vision, conceptualization, experimental design, open sourcing, manu-script writing; N.S.: supervision, experimental design, manuscript writing.

## Competing interests
The authors declare no competing interests.
