## [Peer Review File · Nature Communications]

Extracting medicinal chemistry intuition via preference machine learningREVIEWER COMMENTS

Reviewer #1 (Remarks to the Author):

The manuscript by Choung et al. titled “Learning chemical intuition from humans in the loop” is an interesting study on applying artificial intelligence to extract the latent ranking of molecules based on whether they would be preferred in a drug discovery programme. The authors applied established ranking algorithms to a new setting and were able to bias the results towards (or away from) molecular features that influence the decisions taken by experienced medicinal chemists but are not captured by commonly used scoring functions, such as the QED values, NIBR filters or the Rules of Five. Those results are of great use to any molecular discovery community where large numbers of structures are generated but then the datasets are reduced to human-tractable sizes.

The study is well-structured and the shortcomings of previous reports are clearly highlighted. I particularly liked the study design that tried to account for biases in ranking responses, such as the anchoring and the left/right bias. As a result, inter- and intra-rater agreements were really good for the inherently subjective problem of “which molecule do you prefer?”. I have no major concerns with the study design and methods choice. I have a minor question about data filtering and curation. I really liked how care was taken to remove salts, normalise tautomers and neutralise atoms. While I guess that the decision to remove compounds with more than 10 rotatable bonds is based on their excessive flexibility, I am not sure why those with more than three fused rings were also neglected as many FDA-approved drugs (e.g., steroids, alkaloids, tetracyclines) have many fused rings. I do not think this invalidates the study design or findings in any way, I just think it warrants a side comment on why those decisions were taken.

The manuscript was easy to follow while maintaining a communication style. Technical details of the learning to rank and active learning tasks could be found in the associated GitHub repository. I was also able to easily find the data containing the 5,276 rankings necessary to reproduce the study. However, I had trouble finding the data pertaining to the analysis and interpretation. In particular, I found the discussion on fragment preference useful – many of the desirable and undesirable fragments make intuitive sense. I was hoping to explore that fragment scoring data to see how often certain fragments appear in the FDA-approved drugs but could not find the fragment and their associated scores in the repository. Perhaps it could be documented a bit more so that the interpretation is easier to reproduce and expand.

Along the same vein, I would love to have seen a bit more interpretation of the results summarised in Figure 4. I completely agree with the authors that the very low Pearson’s correlation coefficients suggest an orthogonal perspective that MolSkill scores provide. It is also not surprising that the strongest correlation was found with the QED score, which attempts to capture drug-likeness. What I found

surprising is how (relatively) well-correlated fingerprint density was, as to me that would suggest that more structurally complex molecules would be preferred over the less feature-rich ones. However, that could be just an exemplification of my personal bias towards easier-to-make molecules rather than potentially active drug molecules.

Overall, I greatly enjoyed reading the manuscript. In particular, it was interesting to see how the resulting model captured years of institutional and personal know-how (or “chemical intuition”). Perhaps the title is slightly misleading, as the model acquired such intuition from a linear stream-lined ranking process more so than “in-the-loop” but I will leave this decision to the Editor as this discussion is purely semantical. After all, one could consider the active learning approach to minimise the uncertainty in the latent score differences as some type of a “loop”. I am also not sure if we – as chemists – “learnt” any chemical intuition as a result but the model certainly has captured a significant part of it, which definitely makes it useful. I would love to see it published and I am certain it will be adopted by the many different communities with interests in cheminformatics and machine learning in chemistry.

Reviewer #2 (Remarks to the Author):

Learning chemical intuition from humans in the loop

Oh-Hyeon Choung, Riccardo Vianello, Marwin Segler, Nikolaus Stiefl¹, and José Jiménez-Luna

The authors present a well-written manuscript and report a study where they aim to replicate the lead discovery & optimization process by applying AI for learning-to-rank compounds based on feedback that was obtained from 35 chemists at Novartis over the course of several months. They evaluate the usefulness of the learned tasks such as compound prioritization, motif rationalization, and biased de novo drug design. The developed models and code are made available through a permissive open-source license. This study will be of interest to the drug discovery readership and community but I have serious questions on the appropriateness for a Nature publication and the underlying assumptions made on the discovery process.

This study is reminiscent of a publication from several years back in the Journal of Medicinal Chemistry Assessment of the Consistency of Medicinal Chemists in Reviewing Sets of Compounds <https://pubs.acs.org/doi/10.1021/jm049740z> (which the authors cite). Many of the experiments and conclusions are similar to the previous publication. The advance made here is the AI/ML algorithm that has been developed. So I question the novelty of the study for a Nature publication.

Secondly and more importantly, the days of drug discovery and development where teams of medicinal chemists select compounds based on their chemical intuition is long gone! It is data-driven decisions based on multi-property biological profiles that is the norm in therapeutic area project teams today. The study would have been better or more enhanced (and Nature quality) if AI/ML method were developed based on close-in analogs of particular compounds within a chemical series + the features and chemical desirability (that the authors here have enabled) to discriminate between superior profiles (and thereby compounds) as defined by project teams. Chemical desirability profiles are only one factor in the process.

While this work is worthy of a publication, it lacks the novelty and relevance of what is happening currently in the industry in drug discovery project teams that this is certainly not Nature quality. Better sent to ACS journals such as Journal of Medicinal Chemistry or Journal of Chemical Information and Modeling for review and publication.

Reviewer #3 (Remarks to the Author):

Thank you for the opportunity to review the manuscript "Learning chemical intuition from Humans in the loop" by Choung et al.

The key message of the study

The article (Choung et al.) showed that by using Artificial Intelligence -developed based on human decisions - it is possible to shorten medicinal chemists' work during complex lead optimization. By combining people's real-like tasks for the development of AI in this context, the paper gives a significant contribution to the drug discovery process, human decision-making in complex situations, and the development of human-like AI and their interactions.

Validity

On a technical level, the validity of data collection, interpretation, and conclusion is robust and professional. However, the Choung et al. work is based on the human subjects' pairwise learning-to-rank experimental design. Despite this, the authors argue, that "... making binary decisions is a task that is in general easier.", this simplifying way might also make the decision situation too simple compared to the real work of medicinal chemists. I don't mean that the authors should collect new material, but the quality of the work would be improved if the authors were aware that overly simplistic approaches give

a too narrow and difficult-to-generalize picture of human behavior. It would be good to refer to, for example, Jolly & Cheng (2019), Hasson et al. (2020), or Suomala & Kauttonen (2022) in the introduction and/or discussion to be conscious that it is possible to create more multidimensional experiments today.

In addition, a short comparison of the author's approach to intuitions' role in experts' work compared to studies in cognitive psychology - and decision-making studies is welcome (Again in the Introduction or/and Discussion).

Significance

The manuscript has high potential significance in decision-making among drug discovery campaign contexts. In the future, the authors' approach would be a significant contribution to human decision-making, choice, and behavior in general in complex real-like situations in cognitive science and economics, if researchers in the future change the experiments more real-like [from pairwise learning to complex situation learning; see Jolly & Cheng (2019)].

Data and methodology

Although I have applied artificial intelligence with my colleagues in the study of human decision-making and behavior, my role has been in formulating hypotheses, producing experimental setups and their contents, and interpreting the results. This is because my abilities are not sufficient for technical-level analysis of AI codes. However, on a meta-level, I have an understanding of the steps involved in utilizing and creating AI requires human intelligence to simulate studies. Based on this meta-level expertise of AI my conclusion is that the data and methodology of the manuscript are on an excellent level.

Analytical approach

Statistical tests have been implemented matter-of-factly and professionally.

Suggested improvements

As I wrote above, more analysis on a theoretical level is welcome. Especially about the problems of too simple experiments and the previous studies about the role of intuition in experts' work. We are living in the middle of a replication crisis and one of the causes of this crisis might be too simple experiments. At the very least, researchers should be aware of these problems.

Authors wrote: "One of such advantages is that there is plenty of evidence from psychometric studies and decision theory suggesting that humans find it inherently hard to sort items according to their preferences, whereas making binary decisions is a task that is, in general, easier." However, it would be important for the authors to also highlight the problems such "binary decisions" approach causes in the study of human behavior. It is at least two problems relating to this simplistic approach: the simplistic approach targets research hypotheses to things that are not relevant in people's real decision situations (See, f.ex. Camerer, 2013; Hayden & Niv, 2021; Suomala & Kauttonen, 2023; Yarkoni, 2020). The second problem is that the cognitive bias research tradition (Daniel Kahneman and others) uses this simplistic approach as a benchmark for rationality. Then they found that humans are irrational because they do have not consistent preferences, and use emotions and contextual cues when making decisions.

However, there is growing evidence that human is rational because they have intuitive models, can flexibly behave in different contexts, and uses emotions to achieve personal goals (Gershman, 2019; Suomala & Kauttonen, 2022). According to this “new approach”, cognitive biases are often “rational” and essential properties of the system (artificial or biological), which behave in a complex situation with limited energy (See Gershman, 2021).

This critique of mine is meant to help the writers perfect what is already an excellent manuscript. It would therefore be good for the authors to briefly reflect on the limitations at the theoretical level (in the Introduction/Discussion) caused by simplified experimental setups (which are admittedly still the mainstream in the behavioral sciences).

Clarity and context

See previous comments. One technical detail: You should explain more specific information about Figure 1 in the headline text. Maybe you can divide the Figure into the A-, B- and C-parts and explain the phase-by-phase research process.

Conclusion

I am not an AI expert on a technical level. Thus I can not review the codes of AI. However, I understand the pipeline of this kind of research and on the meta-level the paper is very good and the information has been presented clearly. This is an excellent paper. Some theoretical analyses are welcome (See above).

Reference list of the publications I referred to in my review above. Maybe some of these help authors finalize their manuscript:

Camerer, C. F. (2013). A Review Essay about Foundations of Neuroeconomic Analysis by Paul Glimcher. *Journal of Economic Literature*, 51(4), 4. <https://doi.org/10.1257/jel.51.4.1155>

Gershman, S. J. (2019). How to never be wrong. *Psychonomic Bulletin & Review*, 26(1), 13–28. <https://doi.org/10.3758/s13423-018-1488-8>

Gershman, S., j. (2021). *What makes us smart: The computational logic of human cognition*. Princeton University Press.

Hasson, U., Nastase, S. A., & Goldstein, A. (2020). Direct Fit to Nature: An Evolutionary Perspective on Biological and Artificial Neural Networks. *Neuron*, 105(3), 3. <https://doi.org/10.1016/j.neuron.2019.12.002>

Hayden, B. Y., & Niv, Y. (2021). The case against economic values in the orbitofrontal cortex (or anywhere else in the brain). *Behavioral Neuroscience*, 135(2), 192–201. <https://doi.org/10.1037/bne000448>

Jolly, E., & Chang, L. J. (2019). The Flatland Fallacy: Moving Beyond Low-Dimensional Thinking. *Topics in Cognitive Science*, 11(2), 2. <https://doi.org/10.1111/tops.12404>

Suomala, J., & Kauttonen, J. (2022). Human's Intuitive Mental Models as a Source of Realistic Artificial Intelligence and Engineering. *Frontiers in Psychology, 13*, 873289.
<https://doi.org/10.3389/fpsyg.2022.873289>

Suomala, J., & Kauttonen, J. (2023). Computational meaningfulness as the source of beneficial cognitive biases. *Frontiers in Psychology, 14*, 1189704. <https://doi.org/10.3389/fpsyg.2023.1189704>

Yarkoni, T. (2020). The generalizability crisis. *Behavioral and Brain Sciences, 1–37*.
<https://doi.org/10.1017/S0140525X20001685>

REVIEWER: Jyrki Suomala, Principal Lecturer (Ph.D.)

Laurea University of Applied Sciences

FINLAND

Reviewer #1 (Remarks to the Author):

The manuscript by Choung et al. titled “Learning chemical intuition from humans in the loop” is an interesting study on applying artificial intelligence to extract the latent ranking of molecules based on whether they would be preferred in a drug discovery programme. The authors applied established ranking algorithms to a new setting and were able to bias the results towards (or away from) molecular features that influence the decisions taken by experienced medicinal chemists but are not captured by commonly used scoring functions, such as the QED values, NIBR filters or the Rules of Five. Those results are of great use to any molecular discovery community where large numbers of structures are generated but then the datasets are reduced to human-tractable sizes.

The study is well-structured and the shortcomings of previous reports are clearly highlighted. I particularly liked the study design that tried to account for biases in ranking responses, such as the anchoring and the left/right bias. As a result, inter- and intra-rater agreements were really good for the inherently subjective problem of “which molecule do you prefer?”. I have no major concerns with the study design and methods choice. I have a minor question about data filtering and curation. I really liked how care was taken to remove salts, normalise tautomers and neutralise atoms. While I guess that the decision to remove compounds with more than 10 rotatable bonds is based on their excessive flexibility, I am not sure why those with more than three fused rings were also neglected as many FDA-approved drugs (e.g., steroids, alkaloids, tetracyclines) have many fused rings. I do not think this invalidates the study design or findings in any way, I just think it warrants a side comment on why those decisions were taken.

Thanks for pointing this out, as it was not entirely well explained in the text. We did not remove all compounds with either more than 10 rotatable bonds or more than three fused rings. Instead, a comparison between two compounds was excluded when both of them fulfilled either the fused rings or rotatable bonds restrictions. The main intention behind this was to avoid ‘pest vs. cholera’ comparisons where chemists may reject one compound or the other arbitrarily.

These filtering decisions were made based on informal input obtained from the participating chemists at Novartis on early study rounds. It is correct that some drugs on the market would be removed by these filters, however, in the eye of a medicinal chemist these structures often stand out as not favourable. As the target of the study is to create a scoring function to approximate chemists preferences, we believed that including both molecules with said features on the same pair would not add a lot of valuable information. In addition, these specific compound features can easily be captured by rules and hence can be easily included on a cheminformatics pipeline in more straightforward ways. We however note that the chosen filtering thresholds are somewhat arbitrary and that less stringent limits could have been chosen. In this revision we have revised the description and added a comment about these choices under the Materials section:

“[...] Additionally, pairs with both compounds featuring either more than 10 rotatable bonds or 3 fused rings were removed. This was done to avoid comparisons where chemists may reject one compound or the other without a clear preference for either. We note that these filters were mostly informed by early informal discussions with the participating chemists, but are fairly arbitrary - less stringent boundaries could be chosen for future studies.”

The manuscript was easy to follow while maintaining a communication style. Technical details of the learning to rank and active learning tasks could be found in the associated GitHub repository. I was also able to easily find the data containing the 5,276 rankings necessary to reproduce the study. However, I had trouble finding the data pertaining to the analysis and interpretation. In particular, I found the discussion on fragment preference useful – many of the desirable and undesirable fragments make intuitive sense. I was hoping to explore that fragment scoring data to see how often certain fragments appear in the FDA-approved drugs but could not find the fragment and their associated scores in the repository. Perhaps it could be documented a bit more so that the interpretation is easier to reproduce and expand.

Many thanks for going through the repository and making us aware that some data is missing (it was included in a private branch and we had overlooked to merge it to the main branch) . In this revision we have now updated the main branch of the repository to include fragment data, accompanied by their associated MolSkill scores using the last version of the model. We also now include the biased molecular generation data that was used as base for the analyses on Section 4.6. These are present now under the `data/` directory and a short description has been added to the `data/README.md` file so that details regarding these analysis are not lost on the reader.

Along the same vein, I would love to have seen a bit more interpretation of the results summarised in Figure 4. I completely agree with the authors that the very low Pearson's correlation coefficients suggest an orthogonal perspective that MolSkill scores provide. It is also not surprising that the strongest correlation was found with the QED score, which attempts to capture drug-likeness. What I found surprising is how (relatively) well-correlated fingerprint density was, as to me that would suggest that more structurally complex molecules would be preferred over the less feature-rich ones. However, that could be just an exemplification of my personal bias towards easier-to-make molecules rather than potentially active drug molecules.

In this revision we have extended this section to further incorporate some insights regarding the relationship between MolSkill scores and other in silico metrics. We note however, that since all of the correlations shown in Figure 4 are quite poor ($\sim < .3$), most of the interpretations should be taken with a fair degree of caution. As an additional comment, in our opinion it is not surprising that FP density is correlated to what medicinal chemists like, and we had specifically included it as an interesting feature to follow. Examples of molecular features that would result in low FP density are repeating motifs containing very similar atom types (*e.g.*, repeating aliphatic carbon chains), which could be prone to unspecific binding. This gut feeling, is based on years of experience of following project teams when prioritizing compounds from HTS screening data. Here, feature-poor compounds are often not selected for follow up as it is unclear on how to optimize them or because the team members believe them to be unspecific binders.

"[...] Not surprisingly, the most correlated descriptor in these analyses is QED, which also attempts to capture drug-likeness. Interestingly, the fact that different flavours of fingerprint density are also present within this list suggests that the chemists seem to display a small preference towards richer molecules feature-wise. To some degree this is not surprising, as

one main example of features that would result on low fingerprint density are repeating motifs of similar atom types (e.g., long aliphatic chains), which are difficult to optimize or prone to overall unspecific binding. In contrast, a small positive correlation with the SA score measure (Ertl2009estimation) can be observed, which hints that the proposed score slightly prefers synthetically simpler compounds. Another noteworthy fact is that the SMR VSA3 descriptor, a measure of molecular surface area that is aggregated according to Wildman-Crippen MR value limits (Labute2000widely), is slightly correlated negatively. This could hint that chemists have a liking towards molecules that feature neutral nitrogen atoms. We however stress that the magnitude of these correlations is, in our opinion, insufficient to make any strong claims. However, for completeness, an extensive list of all of the properties computed as well as their correlations to the learned scores is also provided in Figure S3"

Overall, I greatly enjoyed reading the manuscript. In particular, it was interesting to see how the resulting model captured years of institutional and personal know-how (or “chemical intuition”). Perhaps the title is slightly misleading, as the model acquired such intuition from a linear stream-lined ranking process more so than “in-the-loop” but I will leave this decision to the Editor as this discussion is purely semantical. After all, one could consider the active learning approach to minimise the uncertainty in the latent score differences as some type of a “loop”. I am also not sure if we – as chemists – “learnt” any chemical intuition as a result but the model certainly has captured a significant part of it, which definitely makes it useful. I would love to see it published and I am certain it will be adopted by the many different communities with interests in cheminformatics and machine learning in chemistry.

Many thanks for the overall positive feedback, we would like to see the proposed approach adopted alongside other commonly-used metrics in cheminformatics such as QED or SA Score. In regards to how the term “in the loop” is used in the manuscript, we agree that it is mostly a semantic discussion, as we were mostly referring to how the different batches of data were provided to the chemists in an active learning fashion. Still, in our opinion it does also reflect how we are exploring future applications of the method to other endpoints. For example, in a project specific setting, the overall loops are much “smaller” and the score is more iteratively optimized (and may also change over time given latest SAR data). Nevertheless, we would be happy to modify the title if the editor handling the manuscript thinks the used terminology can lead to confusion.

Reviewer #2 (Remarks to the Author):

Learning chemical intuition from humans in the loop

Oh-Hyeon Choung, Riccardo Vianello, Marwin Segler, Nikolaus Stiefl¹, and José Jiménez-Luna

The authors present a well-written manuscript and report a study where they aim to replicate the lead discovery & optimization process by applying AI for learning-to-rank compounds based on feedback that was obtained from 35 chemists at Novartis over the course of several months. They evaluate the usefulness of the learned tasks such as compound prioritization, motif rationalization, and biased de novo drug design. The developed models and code are made available through a permissive open-source license. This study will be of interest to the drug discovery readership and community but I have serious questions on the appropriateness for a Nature publication and the underlying assumptions made on the discovery process.

Many thanks for the positive feedback and for constructively expressing concerns about this manuscript. Please find a point-by-point response to each of the comments below.

This study is reminiscent of a publication from several years back in the Journal of Medicinal Chemistry Assessment of the Consistency of Medicinal Chemists in Reviewing Sets of Compounds <https://pubs.acs.org/doi/10.1021/jm049740z> (which the authors cite). Many of the experiments and conclusions are similar to the previous publication. The advance made here is the AI/ML algorithm that has been developed. So I question the novelty of the study for a Nature publication.

While the mentioned study is indeed related to our work, they are inherently different from each other, address different goals, and reach different conclusions. Novelty-wise, we believe we have made significant contributions in:

Study design and conclusions: The aforementioned study from Lajiness *et al.* tried to measure to what degree medicinal chemists agree at rejecting compounds from a given list. Our study design is inherently different as we focus on pairwise comparisons, which in the psychometric and behavioural psychology literature is a much more robust way of evaluating human preferences. Specifically, the former study suffers from several biases such as anchoring and subject/situation-specific reference values.

This has a clear effect on the conclusions drawn from the studies. From Lajiness *et al.* it was concluded that “[...] *From the results presented here one must conclude that medicinal chemists are not consistent with themselves or compared to other medicinal chemists*”. In the presented study, however, we find that the proposed study design resulted in a much higher agreement, both within, and between medicinal chemists, as shown by the intra and inter-rater agreement coefficients provided in Table 1.

Motivated by the agreement between the raters, in this study we go beyond evaluating whether chemists agree with their choices, but propose the development of a human-learned scoring function that can be used for several relevant cheminformatics tasks. We also exemplify several of these tasks in the manuscript.

ML development: In this study we cast the pairwise psychometric study setup as a learning to rank problem, which infers a score for each compound without ever having chemists provide it explicitly. This has several advantages, the main one being that we are able to explore chemical space in a wide manner without needing to have chemists evaluate the

same pair of compounds in repeated occasions, which was a major limitation on previous studies (Sheridan, 2014).

Open sourcing: We provide an open-sourced package (MolSkill) that can be easily integrated within other cheminformatics pipelines, to make it accessible and usable for the community. In contrast, the referenced study has not even made their data available.

To the best of our knowledge, we believe that we have made several significant contributions in regards to how these studies can be run, and show that it is indeed possible to effectively learn a crowdsourced-version of drug attractiveness using the proposed study design and machine-learning methodology. From in-house experience at Novartis, we are aware that the approach is routinely used to deprioritize *de-novo* generated compounds that are likely not interesting for the user. We believe this is a very important step for acceptance of ML-based generative approaches. In turn, this contributes towards fighting pervasive opinions such as "*(Generative AI) may produce structures, but they are never reflecting what a medicinal chemist would like*" (as expressed by a participating in-house chemist). Several team members have expressed that MolSkill signifies a major step towards making these methods practical in actual drug design

So that these aspects are not lost on the reader, we have further stressed these contributions in several sections of the manuscript:

"[...] Proof-of-concept data collection rounds were carried out to evaluate whether the proposed study design and methodology successfully overcame cognitive bias limitations that were present in previous studies."

"[...] In the specific context of drug discovery applications, it should be observed that this labeling strategy is markedly different from ones used in previous studies (Sheridan2014modeling), where it was found that a large number of repetitions per compound was crucial to obtain a reliable proxy for the crowd-sourced endpoint of interest. This was mostly due to the overall low reported agreement between chemists, as also reported by another studies (Lajiness2004assessment). We partially attribute this difference in agreement to the human anchoring effect when rating items in a sequential manner."

"[...] It is important to note that the proposed methodology is markedly different in design from previous studies, which directly model the human-provided feedback as a regression objective. Our approach, on the other hand, infers these scores automatically based on the preferences expressed, which, due to the study design are more robust between and within chemists. This in practice allows us to prioritize the exploration of a wider chemical space rather than having to rely on repeated feedback from different chemists."

Secondly and more importantly, the days of drug discovery and development where teams of medicinal chemists select compounds based on their chemical intuition is long gone! It is data-driven decisions based on multi-property biological profiles that is the norm in therapeutic area project teams today. The study would have been better or more enhanced (and Nature quality) if AI/ML method were developed based on close-in analogs of particular compounds within a chemical series

+ the features and chemical desirability (that the authors here have enabled) to discriminate between superior profiles (and thereby compounds) as defined by project teams. Chemical desirability profiles are only one factor in the process.

As a team with several decades of industrial drug discovery experience in total, we do agree that drug discovery is an inherently multi-dimensional process where the influence of many (possibly colluding) variables need to be taken into account (e.g., ADMET, specificity, stability). It was not our intention to claim that one can reduce the entire process of drug discovery to the usage of the proposed metric. Each dimension/metric provides a complimentary view on how appropriate a compound is according to specific criteria. Along those lines, we believe that the proposed metric provides an orthogonal view that is currently not captured by existing descriptors (see Section 4.3 for an extended discussion on this, as requested by Reviewer #1).

Tangentially, not only are the days of selecting compounds based on chemical intuition gone, but also those of solely designing compounds based on a medicinal chemist's thought process. As mentioned before, novel methods such as Generative Chemistry, Matched Molecular Series, or full-scale compound/fragment enumerations such as GDB require the access to methods that can handle large sets of compounds with a satisfactory predicted profile but beyond the "usual" ideas a chemist comes up with. In our own experience, enumerations in the millions of compounds along a single R-vector is common these days and while a large extent of compounds will not have a good profile, there are often still millions with a "good" profile that need further filtering and only some of those can be described with existing filters due to their inherent novelty (and consequently, out of the known domain of applicability of existing chemical space).

"[...] Along those same lines, the proposed score can also be used to prioritize combinatorially-generated chemical libraries, whose inherent novelty makes them hard to filter with existing filters."

In the presented study we had mainly focused on training a target-unaware metric that could be used in the same vein as QED or SA Score for prioritization. This had also simplified the overall study setup and question posed.

"[...] We also show that the proposed learned scoring function can better capture the concept of drug-likeness more accurately than another widely used metric (QED)"

How to adapt the proposed approach to a target-specific or series-specific scenario, however, remains a topic of future research, as it would require a completely different study setup, scale, and nuance than the one that is presented here. However, so as to further clarify the difficulties of an actual drug discovery campaign, we have extended the Discussion section to incorporate some of the limitations expressed here.

"[...] Another venue of further research is examining the utility of the study setup in prospective, target-specific lead optimization scenarios, where information from multiple sources (e.g., biological profiles, ADMET) need to be taken into account as a whole to successfully deliver a drug to the market. [...]"

While this work is worthy of a publication, it lacks the novelty and relevance of what is happening currently in the industry in drug discovery project teams that this is certainly not Nature quality.

Better sent to ACS journals such as Journal of Medicinal Chemistry or Journal of Chemical Information and Modeling for review and publication.

We appreciate that the reviewer believes that this work is worthy of publication. However, we disagree on the point of this work not being of enough quality for this particular journal. From a hands-on experience from ongoing MedChem projects at Novartis we do know that MolSkill is being currently used in cutting-edge projects. Specifically, in an era where generative ML algorithms are routinely used to generate tens of thousands of compounds in early stages of a project, the ability to include chemists' intuition without having them to explicitly examine them can substantially accelerate the adoption of generative approaches. We have further stressed this point on the Discussion section of the manuscript.

"[...] Finally, from a hands-on experience from several ongoing medicinal chemistry projects at Novartis, MolSkill is currently being applied in several routine tasks. Specifically, in an era where machine-learning methods can design tens of thousands of compounds, or technologies such as high-throughput screening can highlight a large number of hits at early stages of the drug discovery process, the proposed score is being used to implicitly incorporate chemists' intuition for compound filtering without manual examination. This usage will, hopefully, accelerate both the adoption and trust on generative approaches in the upcoming years."

Reviewer #3 (Remarks to the Author):

Thank you for the opportunity to review the manuscript "Learning chemical intuition from Humans in the loop" by Choung et al.

The key message of the study

The article (Choung et al.) showed that by using Artificial Intelligence -developed based on human decisions - it is possible to shorten medicinal chemists' work during complex lead optimization. By combining people's real-like tasks for the development of AI in this context, the paper gives a significant contribution to the drug discovery process, human decision-making in complex situations, and the development of human-like AI and their interactions.

Validity

On a technical level, the validity of data collection, interpretation, and conclusion is robust and professional. However, the Choung et al. work is based on the human subjects' pairwise learning-to-rank experimental design. Despite this, the authors argue, that " ... making binary decisions is a task that is in general easier.", this simplifying way might also make the decision situation too simple compared to the real work of medicinal chemists. I don't mean that the authors should collect new material, but the quality of the work would be improved if the authors were aware that overly simplistic approaches give a too narrow and difficult-to-generalize picture of human behavior. It would be good to refer to, for example, Jolly & Cheng (2019), Hasson et al. (2020), or Suomala &

Kauttonen (2022) in the introduction and/or discussion to be conscious that it is possible to create more multidimensional experiments today.

In addition, a short comparison of the author's approach to intuitions' role in experts' work compared to studies in cognitive psychology - and decision-making studies is welcome (Again in the Introduction or/and Discussion).

Thank you for your comments about the overall validity of the study. We believe that we have addressed your concerns regarding the chosen study setup to capture human behaviour in the Suggested Improvements section.

Significance

The manuscript has high potential significance in decision-making among drug discovery campaign contexts. In the future, the authors' approach would be a significant contribution to human decision-making, choice, and behavior in general in complex real-like situations in cognitive science and economics, if researchers in the future change the experiments more real-like [from pairwise learning to complex situation learning; see Jolly & Cheng (2019)].

Thank you for pointing out the potential significance of this work in the context of drug discovery. We have included the suggested reference in the Discussion section, alongside a comment on the "Flatland fallacy" that is likely present in high-dimensional choice scenarios, such as this study.

Data and methodology

Although I have applied artificial intelligence with my colleagues in the study of human decision-making and behavior, my role has been in formulating hypotheses, producing experimental setups and their contents, and interpreting the results. This is because my abilities are not sufficient for technical-level analysis of AI codes. However, on a meta-level, I have an understanding of the steps involved in utilizing and creating AI requires human intelligence to simulate studies. Based on this meta-level expertise of AI my conclusion is that the data and methodology of the manuscript are on an excellent level.

Thank you for the positive feedback on this point.

Analytical approach

Statistical tests have been implemented matter-of-factly and professionally.

Thank your checking the statistical correctness of the analyses provided.

Suggested improvements

As I wrote above, more analysis on a theoretical level is welcome. Especially about the problems of too simple experiments and the previous studies about the role of intuition in experts' work. We are living in the middle of a replication crisis and one of the causes of this crisis might be too simple experiments. At the very least, researchers should be aware of these problems.

Authors wrote: “One of such advantages is that there is plenty of evidence from psychometric studies and decision theory suggesting that humans find it inherently hard to sort items according to their preferences, whereas making binary decisions is a task that is, in general, easier.” However, it would be important for the authors to also highlight the problems such “binary decisions” approach causes in the study of human behavior. It is at least two problems relating to this simplistic approach: the simplistic approach targets research hypotheses to things that are not relevant in people's real decision situations (See, f.ex. Camerer, 2013; Hayden & Niv, 2021; Suomala & Kauttonen, 2023; Yarkoni, 2020). The second problem is that the cognitive bias research tradition (Daniel Kahneman and others) uses this simplistic approach as a benchmark for rationality. Then they found that humans are irrational because they do not have consistent preferences, and use emotions and contextual cues when making decisions. However, there is growing evidence that humans are rational because they have intuitive models, can flexibly behave in different contexts, and use emotions to achieve personal goals (Gershman, 2019; Suomala & Kauttonen, 2022). According to this “new approach”, cognitive biases are often “rational” and essential properties of the system (artificial or biological), which behave in a complex situation with limited energy (See Gershman, 2021).

This critique of mine is meant to help the writers perfect what is already an excellent manuscript. It would therefore be good for the authors to briefly reflect on the limitations at the theoretical level (in the Introduction/Discussion) caused by simplified experimental setups (which are admittedly still the mainstream in the behavioral sciences).

Thank you for the suggestions and for the provided references. In this revision we have put some of the outlined limitations of the study design, as well as some of the criticisms regarding the definition of rationality in the Discussion section. Specifically, we now acknowledge that there is a growing body of evidence pointing that the picture surrounding rationality is much more complex than the one painted by traditional behavioural psychology. A comment about the Flatland fallacy, and on how chemists in our study have likely used different simplifications to drive compound selection has also been briefly discussed.

“[...] Additionally, while the proposed study design results in a higher agreement between participants than when compared to previous works, the pairwise comparison approach is not free from criticism. There is a growing body of evidence from behavioural psychology pointing that the cognitive bias research tradition (Kahneman1984choices), which claims that rationality is based on preference consistency, paints an incomplete picture of intelligence. Several authors (Suomala2022human, Gershman2019never) have instead suggested pointed out that humans can flexibly behave in different contexts and can use emotions to achieve personal goals, which could be interpreted as “rational” cognitive biases (Gershman2021makes). It is also well known that humans have a tendency to simplify high-dimensional problems (such as the one proposed here, choosing a molecule out of a pair) into a handful of variables that they can keep track of cognitively (i.e., the so-called Flatland fallacy (Jolly2019flatland)), which likely depend on the chemist. Accounting for factors such as these is, to our knowledge, still an unexplored question in the field.”

Clarity and context

See previous comments. One technical detail: You should explain more specific information about Figure 1 in the headline text. Maybe you can divide the Figure into the A-, B- and C-parts and explain the phase-by-phase research process.

Thank you for the suggestion. In this revision we have modified the figure as suggested, separating the 3 stages of the overall idea behind the study and significantly extending the caption to be more explicit.

“[...] Overall schematic of the main idea behind the study. (a) Molecules are treated as players playing competitive games against each other, with the probability of one winning over the other is provided by feedback supplied by chemists. For this, the chemists are asked to select one or the other depending on a pre-specified question prompt on a web application. (b) An implicit score model is learned based on this feedback. A two-legged feed-forward neural network with fixed weights in each leg is supplied with pairs of molecules featurized with common cheminformatics descriptors. During training, its parameters are minimized via a binary cross-entropy loss that depends on a latent score difference computed on the molecule pair and the feedback as supplied by the chemists. (c) Once trained, scores can be inferred for any arbitrary molecule, which can then be used for downstream cheminformatics tasks.”

Conclusion

I am not an AI expert on a technical level. Thus I can not review the codes of AI. However, I understand the pipeline of this kind of research and on the meta-level the paper is very good and the information has been presented clearly. This is an excellent paper. Some theoretical analyses are welcome (See above).

Thank you for the overall positive and constructive review. Your feedback has provided the presented work more nuance and rigor, as well as a wider perspective on aspects related to behavioural psychology.

Reference list of the publications I referred to in my review above. Maybe some of these help authors finalize their manuscript:

Camerer, C. F. (2013). A Review Essay about Foundations of Neuroeconomic Analysis by Paul Glimcher. *Journal of Economic Literature*, 51(4), 4. <https://doi.org/10.1257/jel.51.4.1155>

Gershman, S. J. (2019). How to never be wrong. *Psychonomic Bulletin & Review*, 26(1), 13–28. <https://doi.org/10.3758/s13423-018-1488-8>

Gershman, S., j. (2021). *What makes us smart: The computational logic of human cognition*. Princeton University Press.

Hasson, U., Nastase, S. A., & Goldstein, A. (2020). Direct Fit to Nature: An Evolutionary Perspective on Biological and Artificial Neural Networks. *Neuron*, 105(3), 3. <https://doi.org/10.1016/j.neuron.2019.12.002>

Hayden, B. Y., & Niv, Y. (2021). The case against economic values in the orbitofrontal cortex (or anywhere else in the brain). *Behavioral Neuroscience*, 135(2), 192–201. <https://doi.org/10.1037/bne0000448>

Jolly, E., & Chang, L. J. (2019). The Flatland Fallacy: Moving Beyond Low–Dimensional Thinking. *Topics in Cognitive Science*, 11(2), 2. <https://doi.org/10.1111/tops.12404>

Suomala, J., & Kauttonen, J. (2022). Human’s Intuitive Mental Models as a Source of Realistic Artificial Intelligence and Engineering. *Frontiers in Psychology*, 13, 873289. <https://doi.org/10.3389/fpsyg.2022.873289>

Suomala, J., & Kauttonen, J. (2023). Computational meaningfulness as the source of beneficial cognitive biases. *Frontiers in Psychology*, 14, 1189704. <https://doi.org/10.3389/fpsyg.2023.1189704>

Yarkoni, T. (2020). The generalizability crisis. *Behavioral and Brain Sciences*, 1–37. <https://doi.org/10.1017/S0140525X20001685>

REVIEWER: Jyrki Suomala, Principal Lecturer (Ph.D.)

Laurea University of Applied Sciences

FINLAND

REVIEWERS' COMMENTS

Reviewer #1 (Remarks to the Author):

The revised manuscript by Choung et al. addresses all my previous comments and clarifies the points that could have been confusing to the reader. Authors' argument about the feature richness in molecules with greater fingerprint density resulting in more specific drug binding make perfect sense and were not obvious to me beforehand. I am very happy to see the inclusion of more data in the accompanying repository and I agree with the authors that the manuscripts value is strengthened by the availability of an open-source package (MolSkill) that can be easily deployed in different pipelines.

I would also like to support Authors' view that the manuscript provides substantial advances that could have otherwise been lost among the readership of more specific journals. Coming from a non-medicinal chemistry background, I strongly believe that I would have probably missed the study (unless brought to my attention by citation alerts) if published in the Journal of Medicinal Chemistry, for example. At the same time, I am very interested in modelling molecular synthesizability and believe it to be an unsolved problem of great importance to a wide range of chemical sciences. The methods presented by Choung et al. are scientifically robust and can be redeployed in other areas of chemistry with relative ease. Another major strength of the manuscript is that MolSkill does not depend on arbitrary metrics (such as the Lipinsky Rule of 5) and indeed tries to capture the intuition of medicinal chemists more holistically. While I agree with Reviewer 2 that it is a complex data-driven multi-property problem, it is nonetheless useful to include human judgement in that complex data array in an automated fashion.

Reviewer #3 (Remarks to the Author):

The authors have taken into account my comments in the manuscript and I have nothing more to point out.

Reviewer #1 (Remarks to the Author):

The revised manuscript by Choung et al. addresses all my previous comments and clarifies the points that could have been confusing to the reader. Authors' argument about the feature richness in molecules with greater fingerprint density resulting in more specific drug binding make perfect sense and were not obvious to me beforehand. I am very happy to see the inclusion of more data in the accompanying repository and I agree with the authors that the manuscript's value is strengthened by the availability of an open-source package (MolSkill) that can be easily deployed in different pipelines.

I would also like to support Authors' view that the manuscript provides substantial advances that could have otherwise been lost among the readership of more specific journals. Coming from a non-medicinal chemistry background, I strongly believe that I would have probably missed the study (unless brought to my attention by citation alerts) if published in the Journal of Medicinal Chemistry, for example. At the same time, I am very interested in modelling molecular synthesizability and believe it to be an unsolved problem of great importance to a wide range of chemical sciences. The methods presented by Choung et al. are scientifically robust and can be redeployed in other areas of chemistry with relative ease. Another major strength of the manuscript is that MolSkill does not depend on arbitrary metrics (such as the Lipinsky Rule of 5) and indeed tries to capture the intuition of medicinal chemists more holistically. While I agree with Reviewer 2 that it is a complex data-driven multi-property problem, it is nonetheless useful to include human judgement in that complex data array in an automated fashion.

Many thanks for your kind words and your valuable feedback and support in the review process. We are excited to see what the community will use the package for / build upon.

Reviewer #3 (Remarks to the Author):

The authors have taken into account my comments in the manuscript and I have nothing more to point out.

We'd like to thank you for your useful feedback in the last revision.